# Numerical Analysis of the Light Modulation by the Frustule of *Gomphonema parvulum*: The Role of Integrated Optical Components

**DOI:** 10.3390/nano13010113

**Published:** 2022-12-26

**Authors:** Mohamed Ghobara, Cathleen Oschatz, Peter Fratzl, Louisa Reissig

**Affiliations:** 1Institute of Experimental Physics, Freie Universität Berlin, Arnimallee 14, 14195 Berlin, Germany; 2Max Planck Institute of Colloids and Interfaces, Department of Biomaterials, Research Campus Golm, Am Mühlenberg 1, 14476 Potsdam, Germany

**Keywords:** pennate diatom frustule, micro-optics, near-field optics, photobiology, finite element method, diffraction-driven focusing, photonic jet, Talbot effect, guided-mode resonance

## Abstract

Siliceous diatom frustules present a huge variety of shapes and nanometric pore patterns. A better understanding of the light modulation by these frustules is required to determine whether or not they might have photobiological roles besides their possible utilization as building blocks in photonic applications. In this study, we propose a novel approach for analyzing the near-field light modulation by small pennate diatom frustules, utilizing the frustule of *Gomphonema parvulum* as a model. Numerical analysis was carried out for the wave propagation across selected 2D cross-sections in a statistically representative 3D model for the valve based on the finite element frequency domain method. The influences of light wavelength (vacuum wavelengths from 300 to 800 nm) and refractive index changes, as well as structural parameters, on the light modulation were investigated and compared to theoretical predictions when possible. The results showed complex interference patterns resulting from the overlay of different optical phenomena, which can be explained by the presence of a few integrated optical components in the valve. Moreover, studies on the complete frustule in an aqueous medium allow the discussion of its possible photobiological relevance. Furthermore, our results may enable the simple screening of unstudied pennate frustules for photonic applications.

## 1. Introduction

In recent years, our attention has been increasingly directed toward biological systems that produce micro- and nanostructured biomaterials that have been optimized—through evolution over billions of years—to find unique solutions to complex physical problems [1,2,3]. The ability of living organisms to produce these materials originates from the power of their cells to manipulate molecules, atoms, and ions through molecular mechanisms occurring at the nanoscale and involving what some define as the engines of creation, i.e., DNA and proteins [4]. Such materials are biosynthesized under ambient conditions in an aqueous solution and do not require large amounts of energy or toxic educts. This is an advantage compared to similar man-made materials, offering many organic and inorganic micro- to nanostructured biomaterials for studies and applications (e.g., [5,6,7,8]). Specifically, inorganic structured biomaterials—obtained via elegant biomineralization processes [9]—are often produced for specific purposes, mainly for mechanical support. The nanostructuring of these materials gives pronounced improvements in the mechanical stability of the skeletal systems compared to bulk material, such as the avoidance of the characteristic brittleness of calcite crystals in the shells of Coccoliths [10]. 

An impressive class of microorganisms that produces nanostructured silica is Bacillariophyceae (i.e., diatoms) [11]. It is considered the most diverse microalgal class globally and exists in almost all aquatic ecosystems [12]. Diatoms exhibit a cell wall of amorphous hydrated silica (the so-called frustule) with species-specific pore patterning often of high regularity [11,13]. These frustules are synthesized and shaped through complex biomineralization processes occurring during the growth and reproduction of their living cells [9,14]. The frustule consists of two valves (epivalve and hypovalve) and a few girdle bands surrounding the cell and connecting the two valves. Based on the frustule symmetry, we differentiate between two main groups: the centrics (of radial symmetry) and the pennates (of bilateral symmetry) [11]. Pennate diatoms can be further divided according to the presence of a particular structure called a “raphe” (i.e., a slit within diatom valves helping living cells in their gliding motility on surfaces by controlling the mucilage secretions [15]) into three main categories: biraphid, monoraphid, and araphid [11]. 

Whilst it is clear that diatom frustules serve as protection with enhanced strength against predators and mechanical stresses thanks to their nanostructuring [16,17,18], other benefits are also linked to their design. The frustules may serve as a sieve membrane, enabling size-dependent permeability of nutrients and other beneficial particles to their living cells while excluding harmful particles, such as viruses [19,20]. In addition to this, their ability to control the diffusion of these molecules across the valves’ surface microtopology has been reported [21]. Furthermore, it has been frequently suggested that the ordered nanopores within the amorphous biosilica might influence light propagation, depending on its frequency, due to their similarity to photonic crystals. This brings up more questions regarding whether the possible photobiological roles of the frustules are an additional reason for their evolutionary success [22,23,24]. Additionally, such nanostructuring makes frustules a promising material for several applications, such as in drug delivery systems, sensors, solar cells, batteries, and many more [8,25,26].

During the last two decades, light propagation across a number of diatom valves belonging to a few species was investigated by employing theoretical and experimental approaches, and these efforts have been reviewed [22,24,27]. These studies demonstrated the valves’ ability to influence the propagation of light through, for instance, focusing [28,29,30], orientation-dependent [31,32] and wavelength-dependent transmission [33], waveguiding [34,35], guided-mode resonance [36], and diffraction grating behavior in the far field [37,38], in addition to their intrinsic photoluminescence properties [39,40]. Recently, light interaction with a girdle band has also been reported, showing a photonic pseudogap associated with its photonic crystal structure [41]. Most of these observations are more pronounced for separated clean valves or girdle bands immersed in air than those immersed in water due to the higher refractive index contrast in air. Nevertheless, some light modulation abilities have been linked to probable photobiological roles of the intact frustule immersed in water. For example, the ability of the frustules of some species to enhance the photosynthesis process by concentrating and redistributing the photosynthetically active radiation (PAR) inside the living cells near the chloroplasts has been suggested in addition to their ability to attenuate the harmful radiation (UV range) or higher light intensities as possible [22,23,24,42]. 

So far, most studies have focused on large-size centric diatom species, for example, *Coscinodiscus* spp., which allows simple experimental conditions due to its size (diameters up to 500 µm [43]). However, it is difficult to make general assumptions about valves’ light modulation capabilities or frustules’ photobiological roles that are valid for the huge number of diatom species (there might be up to 10^5^ species), which have considerable variation in frustule shape and ultrastructure, especially since many diatoms have a small size with different pore patterns, and many species belong to the group of pennates, as indicated by the AlgaeBase website [44]. Recently, the pennate species received some attention through the investigation of the light modulation by the araphid pennate frustule of *Ctenophora pulchella* [29]. It is worth mentioning that some application studies considered pennate diatom valves; however, they were often more concerned with a specific feature (such as photoluminescence or guided-mode resonance) and did not analyze all their light modulation capabilities [36,45]. 

Since detailed experimental investigations of diatom frustules smaller than 30 μm are difficult, numerical analysis of light modulation by these frustules might be particularly promising. Using numerical methods based on geometrical optics assumptions (such as in [46]) or strong approximations may exclude diffraction and interference or lead to inaccurate predictions, respectively. For relatively large valves in the range of a few tens to a few hundreds of µm, the beam propagation method (BPM) has been applied successfully in [28,30,47,48,49]. Additionally, finite difference and finite element methods (FDM and FEM, respectively) are two common numerical approaches that have also been extensively used to solve optical problems related to diatoms [36,41,50,51,52,53]. FD methods, such as the finite difference time domain (FDTD) method, are easy to handle but more suitable for regular structures, as discretization cannot deal efficiently with irregular shapes or edges. On the other hand, FEM—used in this study—has shown advantages as a powerful technique suitable for dealing with arbitrary geometries and even inhomogeneous media [54]. 

In this paper, we aim to expand the knowledge on the light modulation capabilities of diatom frustules by focusing on small pennates, which form a large and diverse group often underrated in such studies. The frustule of biraphid pennate *Gomophonema parvulum* (*G. parvulum*, GP) is used as a model, extending and explaining our preliminary results reported in [55] by employing numerical analysis using frequency domain FEM. To acquire the necessary 3D structural information for building a realistic and statistically representative model for simulations, we use a focus ion beam-scanning electron microscopy (FIB-SEM) workflow in addition to regular SEM analysis. Moreover, we use a novel analytical approach to enhance our understanding and minimize computational costs by (i) reducing the complex 3D structure (the complete valve or frustule) into 2D cross-sections and, further, by (ii) disassembling the distinct optical components. This approach helps us to understand the overlapping optical phenomena and reveals the role of integrated optical components within the solar spectrum range. Furthermore, by investigating the influence of different structural parameters (using analytical models) as well as refractive index contrast on the observed phenomena, this study opens the door for predicting the light modulation by other pennate frustules of similar structure but different dimensions. 

## 2. Materials and methods

The GP structural parameters—necessary for the construction of a representative 3D model and for understanding the variability of distinct structural parameters between valves—were extracted from SEM and FIB-SEM data. For this, GP diatoms (obtained from Goettingen, Germany) were cultivated in 50 ml culture flasks (Greiner Bio-One GmbH) in Wright’s cryptophyte medium [56] in a cultivation cabinet (Percival CFL LED) at 18 °C and a day/night light cycle of 12 h/12 h. The progress of cultivation was observed by an inverted light microscope (Zeiss, Axio Vert.A1). 

### 2.1. Characterization with SEM

SEM micrographs of six different GP valves (two on the internal and four on the external view) were used to obtain structural information in the 2D plane. For this, GP frustules were cleaned from organic components using an acid cleaning procedure. In detail, 8 mL of the diatom culture was mixed with 10 mL of concentrated HNO_3_ in a wide beaker and stirred for 3 h at 65 °C on a hot plate. After cooling, the resulting silica was centrifuged at 8500 rpm and washed three times with deionized water. After extraction, the silica material was freeze-dried until further use. For SEM analysis, several droplets of GP valves’ aqueous suspension were placed on an alumina sample holder. SEM measurements were performed using a JSM-6060LV scanning electron microscope (JEOL) with 5 kV acceleration voltage (*EHT*) and a probe current *I_p_* of 8 nA using a secondary electron detector.

### 2.2. Characterization Using FIB-SEM 

FIB-SEM data were used to provide 3D structural information of GP frustules. In total, ten frustules were analyzed. For FIB-SEM measurements, frustules need to be fixated. In the case of GP diatoms, fixation of cleaned frustules did not lead to desirable image quality due to an agglomeration of the individual valves and girdle bands which hampered the segmentation of the cell wall, and, thus, the fixation was carried out on concentrated (by centrifugation at 200 rpm) intact diatom cells using high-pressure freezing [57,58] and freeze substitution [59]. For this, copper planchettes (typ B) were filled with 5 µl cell suspension and closed with a second planchette. The sample was then loaded into a sample holder and injected into a high-pressure freezing system (Leica EM HPM 100). From that time, the samples were handled under liquid nitrogen to prevent unfreezing. The samples were transferred into a cryo vial containing reagents for cryo substitution (2% osmium tetraoxide + 0.1% uranyl acetate + 0.5% glutaraldehyde + 1.5% water in acetone), and an automatic freeze substitution in a Leica EM AFS 2 was started (starting at −120 °C and slowly heating to 4 °C). After freeze substitution, the planchettes were removed from the sample, and the sample was washed five times with acetone. Then, the fixed GP cells were embedded using a slow infiltration procedure over four days [60] in Epon 812 resin (4 g resin, 2.67 g DDSA (dodenyl succinic anhydride), 1.67 g NMA (nadic methyl anhydride), and 0.168 g BDMA (N,N-dimethylbenzylamine)), and were continuously stirred after mixing. The stepwise infiltration procedure started from 1, 2, 3, and 4 droplets of Epon 812/ ml acetone for 2 h each on the first day, 20%, 30%, 40%, 50%, and 60% resin in acetone for 1.5 h each on the second day, and 80% for 4 h on the third day. In the end, 100% resin was used for infiltration for 24 h. Afterward, the samples were filled into small plastic cups, and the resin was hardened in an oven at 65 °C for 36 h. After hardening, the sample blocks were polished, and an area for acquisition containing GP diatoms was located using SEM.

FIB-SEM was performed on a Crossbeam 540 (Zeiss) equipped with a gallium ion source and secondary electron and backscatter detectors. The sample was mounted on a sample holder and tilted to 54° for milling, polishing, and imaging. First, a trench was made underneath the sample with a beam current of 65 nA, before polishing the surface of the hole with a beam current of 7 nA. Imaging (*EHT* = 2 kV, *I_p_* = 700 pA, working distance of 5.2 mm) was acquired in serial slicing mode with a slice thickness of 31.5 nm corresponding to the distance between the consecutive images in the obtained image stack. The resulting image pixel size was 24.83 nm x 24.83 nm at a magnification of 4.54 kX. 

After the acquisition, the obtained image stack was aligned (registered) and denoised by home-written Python programs using OpenCV (v. 2.4.7), taking advantage of the Fourier shift theorem. The displacement vector between consecutive images was calculated using the *phaseCorrelate* function and then expressed as a shift with respect to the first image. In many cases, curtaining artifacts, giving rise to vertical contrast modulations (stripes), were visible in the stacks, which were corrected following the approach of Münch et al. [61]. For this, correction parameters (such as the depth of the wavelet transform, the wavelet family, and the width of the Gaussian filter) were optimized to visually obtain a corrected stack with the least contrast and information loss and the largest stripe removal. Finally, to improve the signal-to-noise ratio, the image stacks were denoised using the *skimage.restoration* local denoising filters of the scikit-image Python library (v. 0.14.0). Again, the filtering algorithms and parameters were chosen by visually optimizing the corrected images to obtain the largest noise removal with the lowest information loss. 

In the corrected image stack, the silica cell wall was segmented from the cell organelles using the Amira segmentation editor. The wizard wand tool could be used due to the high-contrast differences between the silica and the surrounding resin (Figure 1c). The segmented cell wall was cropped into single valves and converted into a surface. The surface was remeshed to obtain a more regular triangulation and optimized (aspect ratio, dihedral angle, tetrahedral angle) to generate an optimized tetrahedral grid. 

### 2.3. Statistical 3D Model of a GP Valve for Numerical Analysis 

To analyze the role of structural features in GP valves in a methodical fashion, a statistical model of a GP valve was created. This 3D model was constructed using the geometry building tools in COMSOL Multiphysics®. The required structural parameters (see further Table 1) needed for the construction of the 3D model were extracted from both 2D micrographs and 3D reconstructed valves using ImageJ software [62] by averaging a number of measurements in each image *X_av,i_* and calculating the corresponding standard deviation *dX_av,i_*. To obtain not only the variation of the parameter within one valve but also across different valves, the weighted mean *X_w_* of the parameter was calculated, including its internal and external errors, *dX_int_* and *dX_ext_*, respectively. If *dX_int_* was larger than *dX_ext_*, the variation within the valves (within *dX_av,i_*) was larger than the variation between valves and vice versa. This analysis helped the estimation of the significance of the structural precision with the variation seen in the simulation results upon changes in the corresponding structural parameter.

The numerical calculations were performed on 2D cross-sections to allow greater structural variability and reduce the complexity in the obtained results, as well as reduce simulation time. For this, representative 2D longitudinal (CS_long,i_) and vertical (CS_ver,i_) cross-sections were extracted from the 3D model (Appendix A). At this point, the cross-sectioning across the 3D model produced sharp edges, which were smoothed to resemble the reconstructed data from FIB-SEM, as shown in Appendix A.

### 2.4. Numerical Analysis of the Wave Propagation across the GP Cross-Sections 

Frequency domain FEM modeling of the wave propagation in the near field was performed using the wave optics module in COMSOL Multiphysics® 5.5, inspired by the procedures in a COMSOL application note [63]. Unless otherwise stated, each 2D CS was placed into a rectangular simulation box (100 µm height (y-axis) and 40 µm width (x-axis)) at a distance of 4 µm from the input boundary, which was illuminated with a transverse plane wave of 80 µm size and an electric field strength *E_input_* of 1 V/m. The scattering boundary condition was applied to the input boundary on the left, while the remaining boundaries were set as perfectly matched layers (PMLs) to avoid nonphysical reflections. This configuration was chosen after optimization. Unless otherwise stated, the calculations were performed as parametric sweeps, changing the vacuum wavelength of the input wave *λ_vacc_* from 300 nm to 800 nm in 50 nm steps to cover the main radiation of the solar spectrum. Across all wavelengths, the refractive index of the amorphous silica constituting the GP valve *n_v_* was set to 1.46 [64], while the refractive index of the surrounding medium *n_m_* was set to *n_a_* = 1.00 or *n_w_* = 1.33, representing air or water, respectively. The physics-controlled mesh size was, in all cases, much smaller than *λ_vacc_* (reaching a few nm) and automatically adapted to the complexity of the geometry of the CSs. To study the influence of the refractive index contrast, parametric sweeps of 1 < *n_m_* < 1.46 and 1 < *n_v_* < 1.9 were performed.

Moreover, the results were compared to simulation results of 2D analytical models of distinct optical components, such as thin slabs, as well as lens-like, grid-like, and fiber-like structures of silica, under identical conditions (vide infra). This allowed more detailed study of the effect of distinct structural parameters (e.g., length, thickness, striae spacing, and areolae diameter) on the light interference patterns. Furthermore, to understand the relevance of the observed optical phenomena of the valve’s CSs to the photobiology of living cells, the effect of adding 4 girdle bands and a hypovalve to selected CSs was studied in water. 

After computation, unless otherwise stated, two-dimensional images displaying the distribution of the normalized electric field strength *E_Norm_* within the simulation domain were obtained, with an intensity scale (0 to 2 V/m) depicted with a color code ranging from blue (*E_Norm_* < *E_input_*) to white (*E_Norm_* = *E_input_*) to red (*E_Norm_* > *E_input_*). Where necessary, the *E_Norm_* values were extracted by creating 2D cutlines at precise (x, y) positions in the simulation domain.

## 3. Results

### 3.1. Structural Analysis of the GP Frustule

GP is a benthic asymmetric biraphid pennate species. It is widely distributed in various aquatic ecosystems, mainly freshwater ecosystems, and has several varieties that differ, to some extent, in shape and size [65,66]. The frustules of the studied GP diatom (Figure 1, Appendix A, and Table 1) are of elliptic to ovate shape (length *L_v_* = 7.1 ± 0.2 µm, width *W_v_* = 4.59 ± 0.07 µm) consistent with the previous structural description of some GP strains reported in [65]. The two valves, epivalve and hypovalve, have a face of thickness *D_v_* = 0.17 ± 0.01 µm, a curved mantle (i.e., an elevated edge of the valve) of height *h_M_* = 0.58 ± 0.08 µm and width *W_M_* = 0.184 ± 0.009 µm, and are connected by girdle bands (approximately four). The face of each valve is divided by a raphe slit (length *L_ra_* = 5.8 ± 0.2 µm, width *W_ra_* = 0.023 ± 0.004 µm), which lies in a thickened area along the apical axis (i.e., long axis) of the valve called the sternum with a maximum thickness *D_S_* = 0.26 ± 0.03 µm and a half-width ⅟₂*W_S_* = 0.32 ± 0.02 µm. The raphe slit is interrupted at the zone of the central nodule, dividing it into two slits with a spacing *d_ra_* = 0.54 ± 0.06 µm. The nodule (*L_nod_* = 1.568 ± 0.002 µm, *W_nod_* = 0.86 ± 0.03 µm), which is not placed precisely in the center along the apical axis but shifted by about 0.18 µm towards one side, is a dome-shaped area that appears in the internal valve face and reaches a maximum thickness of *D_nod_* = 0.38 ± 0.02 µm. At the valve apexes, where the raphe slits end, the sternum slightly increases its thickness at the internal face, sometimes merging with the mantle, which might be associated with the so-called helictoglossa [11]. On both sides of the sternum (or nodule), rows of punctate areolae (i.e., pores) with a spacing of *d_a_* = 0.214 ± 0.008 µm, so-called striae, extend towards the mantle. The striae occur after the sternum or the nodule except for a single stria at the nodule zone shortened by 1 µm that gives the valve, along with the shifted position of the nodule, the asymmetry. The striae are slightly bent or tilted (see Figure 1) with an average striae spacing expanding from *d_str,min_* = 0.49 ± 0.03 µm to *d_str,max_* = 0.57 ± 0.02 µm, resulting in about 13 visible striae per valve. The areolae are visibly smaller on the external face of the valve compared to the internal face, with diameters of 2*r_a,ext_* = 0.100 ± 0.007 µm and 2*r_a,int_* = 0.15 ± 0.01 µm, respectively. The areolae are further covered with so-called flab-like pore occlusions (of a predicted thickness of *D_occ_* ≈ 0.02 µm), leaving a crescent-like slit, reaching a width of *W_occ_* = 0.017 ± 0.002 µm.

Statistical analysis shows that all studied valves are of comparable dimension, with structural parameters varying by less than 10%. The only exceptions (with deviations up to 17%) are the thickness of the sternum *D_s_*, as well as the width *W_ra_* and spacing *d_ra_* of the raphe slit, the mantle height *h_M_*, and the width of pore occlusion slit *W_occ_*. With the exception of *D_s_* and *W_occ_*, these parameters display dominant *dX_ext_*, indicating a relatively large variation between valves. This is also the case for the thickness of the nodule zone *D_nod_*, the valve length *L_v_* and width *W_v_*, and raphe slit length *L_ra_* but with a *dX_ext_* up to 5%. Interestingly many of these parameters are found not to influence the obtained interference patterns significantly. In contrast, structural parameters describing the dimensions and spacing of the areolae (2*r_a,ext_*, 2*r_a,int_*, and *d_a_*), striae spacing (*d_str_*), and the thickness of the valve *D_v_* display a comparably small variation between valves, evident by their dominant *dX_int_* of up to 7%. These are the parameters that can dramatically change the interference patterns in the simulations (vide infra). 

The fine structure of the girdle bands is not studied in detail, as they are comparably simple and do not contain structural features relevant to the light propagation apart from their width, height, and spacing, which are estimated as *W_girdle_* = *W_M_*, *H_girdle_* = 2.84 µm, and *d_girdle_* = 10–50 nm, respectively. This is in contrast to the girdle bands of some larger species, such as *Coscinodiscus* spp., in which their porous structure dramatically influences the light propagation [41] and, thus, has to be considered during simulations of the whole frustule. 

It should be noted that finer structural features, such as the undulations on the silica or apical pore field, are also not considered in this study, as these do not fall into the length scales close to the studied *λ_vacc_* range and are assumed not to be a determinant to the obtained near-field interference patterns. In general, FIB-SEM can deliver images of a resolution as low as 4 nm [67,68], and, if combined with AFM imaging [69], such structural information could be added in future studies.

Furthermore, the content of elements in the silica (available through energy-dispersive X-ray (EDX) mapping analysis) should be considered in the future, as additives can lead to spatial changes in *n_v_(x, y, z)* similar to those that have been reported for some pennate valves [64]. Changes in *n_v_* could significantly influence light propagation, especially in a low-contrast medium. However, the general trends and features seen here should also then be relevant.

### 3.2. Near-Field Simulation of the 2D Cross-Sections—The Role of Optical Components

The near-field light interaction of a number of representative vertical (five in total) and longitudinal (eight in total) CSs—displayed in Appendix A—is studied. All studied CSs show structural features in the length scales of visible light and induce an interference pattern in the near field, as is evidenced by the red and blue areas (e.g., Appendix A). It is evident that distinct structural features in the CSs induce a specific contribution to the interference pattern. Patterns of structurally complex CSs can be explained by the addition of interference patterns of “their distinct structural components” (see, e.g., Appendix A), such as slab-like, lens-like, grid-like, and fiber-like structures. The near-field interference patterns of such components can often be predicted by theory, e.g., the thin-film interference of thin slabs or guided-mode resonance of grid-like structures (vide infra). Therefore, we separately study a range of optical phenomena occurring in the CSs featuring these specific structural components. It should be noted that the presence or absence of pore occlusions in CSs with a grid-like structure do not have a significant effect on the near-field interference pattern either in the longitudinal or vertical CSs (see, e.g., Appendix A). Therefore, pore occlusions are not considered in further discussions.

The idea of “building” the CSs through the addition of optical components is sketched in Figure 2 (see also Appendix A). The simplest form of a longitudinal cross-section (CS_long,5_ or CS_long,7_, which differ in length) is similar to a thin slab of corresponding thickness (A in Figure 2) with curved and extended edges (B in Figure 2). Slicing the valve across the areolae of consecutive striae leads to the addition of a grid-like structure with spacing *d_str_* (CS_long,4_ or CS_long,6_, which differ in length (grid units) and striae spacing *d_str_*). The presence of 1 µm shortened stria on one side of the valve leads to a defect in the grid-like structure that appears in CS_long,4*_. It should be noted that the areas between the areolae have a plano-convex lens-like structure (C in Figure 2) corresponding to the shape of the costae (i.e., the rips). When approaching the center of the valve, the grid is interrupted by the presence of the nodule zone, adding a larger plano-convex lens-like structure (D in Figure 2) slightly off-center to the grid (CS_long,3_). As soon as the sternum zone is approached, the grid-like structure disappears, but the overall thickness of the CS increases (CS_long,2_, see Appendix A). Slicing directly along the apical axis, the slab-like structure is further cut by the raphe slits, leaving a CS featuring only a lens-like structure in the center and two curved edges (CS_long,1_, Appendix A). 

In the case of vertical CSs, the thin-slab elements with curved elongated edges (similar to, e.g., CS_long,5_) are divided at the center either by adding the raphe (with or without its slit, E_1_ or E_2_ in Figure 2, respectively) or the slightly thicker nodule zone (F in Figure 2) of a triangular-like structure (CS_ver,4_/CS_ver,5_ or CS_ver,2_, respectively). It should be noted that the raphe slit in these structures, which only leads to an interruption of around 23 nm between the two parts, does not lead to significant changes in the near-field interference pattern. In all cases, the thin-slab area in the vertical CSs on both sides of the raphe or nodule can be further divided by a grid-like structure (G in Figure 2) with spacing *d_a_* of varying length (CS_ver,3_ or CS_ver,1_, respectively) corresponding to the slicing of the areolae within one stria.

It should be noted that the width of the curved and elongated edge varies slightly between the CSs depending on the position of slicing within the mantle (0.184 µm ≤ *W_M,CS_* ≤ 0.334 µm). Adding girdle bands to both sides of the CSs (while building 2D CSs across the complete frustule) further elongates the edge by adding these fiber-like structures (H in Figure 2). 

In general, the existence of the thin-slab elements leads to the occurrence of two distinct interferences: (I) thin-film interference and (II) edge diffraction, overlaying in the transmittance region. The existence of fiber-like structures—the mantle or, additionally, the girdle bands in the complete frustule—leads to (III) waveguiding behavior, which affects the corresponding edge diffraction pattern. Moreover, the finite size of the CSs, or the presence of thickened protruding structures associated with the cutting of the nodule zone or the sternum, results in increased interference between the transmitted and the diffracted waves from two (or more) edges. This leads to additional phenomena: (IV) diffraction-driven focusing and, further, (V) photonic jet generation in the transmittance region. Furthermore, the grid-like structures lead to: (VI) a characteristic diffraction grating behavior as well as (VII) guided-mode resonance, which leads to dramatic changes in the interference pattern at a specific range of wavelengths. 

### 3.3. The Observed Optical Phenomena: Description and Analysis 

All cross-sections can modulate light effectively, and generally, the modulation strength is strongly dependent on *λ_vacc_* and *∆n*. As mentioned, a number of distinct optical phenomena are observed and correlated to the optical components in the CSs. Here, these phenomena are separately demonstrated, accompanied by the theoretical expectation, elucidating the role of the corresponding structural parameters where necessary.

#### 3.3.1. Thin-Film Interference

Thin-film interference results either from the interference of the reflected light at the first and second interface between a (with respect to *λ*) thin slab (of thickness *D_sl_*) embedded in a homogeneous medium or the interference of the transmitted light, with a light wave being initially reflected at both internal interfaces, which occurs before leaving the thin slab [70,71]. The light waves interfere maximally constructively or destructively if their difference in path length *Δx* corresponds to a multiple (*N*) of the wave’s wavelength in the slab *N*λ_sl_* or (*1/2 + N*)**λ_sl_*, respectively. This path length corresponds to the distance traveled within the slab, i.e., 2**D_sl_*. Furthermore, in the case of the reflected light, a phase shift of *π* needs to be added [71], as one of the reflections happens at a boundary going from an optically thinner to an optically thicker medium. This leads to constructive interference of the reflected light (i.e., an increase in intensity), being concurrent with destructive interference of the transmitted light (i.e., a decrease in intensity) and vice versa (see Figure 14E in [71]). The wavelength *λ_sl_*, at which either maximally constructive or destructive interference occurs, can be extracted from the consideration above and related to the vacuum wavelength of the incident light *λ_vacc_* using the refractive index of the thin-slab element *n_sl_* (using *λ_vacc_* = *λ_sl_***n_sl_*). Through this relation, the positions of the interference maxima (in relation to *λ_vacc_*) also change with *n_sl_* in addition to *D_sl_* (vide infra). Using Fresnel equations [72,73], the theoretical dependency of the intensity value on *λ_vacc_* for slabs of *D_sl_* and *n_sl_* can be estimated (see dashed line in Figure 3 as well as Appendix A). 

As can be seen in Figure 3c, the *λ_vacc_* associated with the maximal constructive interference shifts towards a longer wavelength by increasing *D_sl_* (shown for *n_sl_* = *n_v_* = 1.46) with only negligible changes in the maximal or minimal intensity values. This shift leads to the observation of more oscillations in the case of thicker slabs in our region of interest; as in all cases, the oscillations increase in density with decreasing *λ_vacc_*. For *D_sl_* = *D_v_* = 170 nm, the intensity of the reflected light is maximal at *λ_vacc_* ≈ 330 nm (and approaching the second maximum in the NIR), while destructive interference occurs at *λ_vacc_* ≈ 495 nm (Figure 3a). This means that the amount of light, which transmits through the valve, is attenuated for UV wavelengths, while the green wavelengths are largely unaffected by thin-film interference effects. In contrast, the interference pattern of the thin-slab element of the sternum (*D_s_* = 260 nm) shows an attenuated transmission at *λ_vacc_* ≈ 305 nm and 505 nm, while the element at the nodule zone (*D_nod_* = 380 nm) mostly affects transmission around *λ_vacc_* ≈ 320 nm, 445 nm, and 740 nm (Figure 3a).

As illustrated in Figure 3a, the *λ_vacc_* dependency of the reflectance and transmittance intensity of the CSs with thin-slab elements is in good agreement with the discussed trend. However, it should be noted that the extraction of the data points from the complex interference pattern is not trivial. In the case of reflectance, the formation of a seemingly standing wave in the *E_Norm_* presentation is caused by the interference of the reflected and the incoming waves (see Figure 3b). This makes the *E_Norm_* value strongly position dependent. To overcome this problem, as well as to minimize the edge diffraction effects, we follow the strategy illustrated in Appendix A. Moreover, as the theoretical calculations expect the intensity of the reflectance and transmission, squaring the extracted *E_Norm_* values is required to obtain a match between both the shape and magnitude of the reflectance and transmission spectra (Appendix A, respectively). It should be noted that—using the method described in Appendix A—*E_Norm,R_* and *E_Norm,T_* values associated with the sternum of CS_long,2_ are extractable, while this is not the case for the nodule zone due to its finite length (*L_nod_*). Despite this, the expected constructive and destructive maxima of the nodule zone are observed in the simulation results. 

Neither the positions of the maxima and minima in the interference pattern nor its shape depend on the refractive index of the medium *n_m_* surrounding the thin slab when calculated relative to *λ_vacc_* (Figure 4b, left). However, the obtained magnitude of the reflected light or the total attenuation of the transmitted light at a specific *λ_vacc_* decreases quadratically when *n_m_* approaches *n_v_* (Figure 4a, left). Thus, the observed effect is strongly attenuated in water when compared to air. 

In contrast to *n_m_*, the refractive index of the thin-slab element *n_v_* strongly influences the position of the spectrum (by changing *λ_sl_* and, thereby, the ratio 2*D_sl_*/*λ_sl_*), leading to a red shift in the spectra, as seen in Figure 4 (right), similar to an increase in *D_sl_* but with a simultaneous rise in magnitude. 

#### 3.3.2. Edge Diffraction

Unlike macroscopic objects, where the light diffraction at the edges is neglectable, edge diffraction becomes significant and dominates the light modulation behavior of microscopic objects of a size comparable to *λ_vacc_* [74]. For an optically transparent thin slab, the characteristic pattern of edge diffraction results from the interference of the secondary wavelet generated at the edge—by being a point source, as can be elucidated by the Huygens–Fresnel principle—and the incident wavefront above the edge or the transmitted wave through the thin slab [75], where, in both cases, there is a spatial phase difference *ΔΦ* that can be explained according to Fresnel–Kirchhoff diffraction [71,74]. This pattern includes bright fringes alternating with dark fringes, which appear in our simulations—on the y-axis—above and below the edge, as in the case of the thin slab of length *L_sl_* = 20 µm, *D_sl_* = *D_v_*, and *n_sl_* = *n_v_* (Figure 5a).

By reducing *L_sl_* (as in the CSs), the secondary wavelets produced at the two opposite edges increasingly interfere with each other inside the transmittance region, as can be seen in S_ana,long5_ (i.e., a thin slab equivalent to CS_long,5_ but with straight edges (Figure 5b)), CS_long,5_ (Figure 5c), and CS_long,7_ (Figure 5d). This disturbs the characteristic interference fringes of the edge diffraction, leading to a diffraction-driven focusing and, further, a photonic jet generation that is discussed separately (vide infra).

Moreover, unlike the diffraction that occurs at straight edges as in S_ana,long5_, the edge diffraction in CSs involves curved edges, i.e., the mantle, where the interference pattern includes countless secondary wavelets produced from infinitesimal points at the edge part fronting the incident wave [76]. Further, due to the tilt of the mantle with respect to the thin-slab element (*θ_M_* ≈ 70°, Appendix A), a portion of the diffracted wave deflects at the edge (indicated with a green arrow in Appendix A), which is significant at shorter *λ_vacc_*. By rotating a thin-slab equivalent to the mantle dimensions (S_ana,M_), a similar trend is observed in a tilted position (the same tilt as the mantle) if compared to the in-plane alignment (Appendix A vs. Appendix A, respectively). 

Altogether, the mantle geometry and its tilt modulate the CSs’ edge diffraction fringes, which may involve waveguiding behavior (vide infra) that leads to changes in *ΔΦ* and can be changed by changing the edge geometrical parameters. This becomes evident when comparing the bright fringes outside and inside the transmittance region of CSs, if compared to those of S_ana,long5_. This modulation includes: (i) increasing *E_Norm_* amplitude of some fringes while decreasing others (Appendix A vs. Appendix A, and also Appendix A), (ii) the change of *E_Norm,in_*/*E_Norm,out_* ratio from ≥ 1 in S_ana,long5_ to < 1 in the CSs (Appendix A), (iii) a spatial delay of the inside diffraction fringes (clear for the first and second fringes in Appendix A vs. Appendix A), and (iv) *λ_vacc_* dependency of the shape and intensity of the first diffraction fringe inside (Appendix A). In all cases, the *E_Norm_* of diffraction fringes decreases with increasing *λ_vacc_* (Appendix A), as expected from the Fresnel–Kirchhoff integral [71].

#### 3.3.3. Waveguiding through Fiber-like Components

The contribution of the fiber-like components (the mantle or girdle bands) to the interference pattern of the CSs is not straightforward due to their waveguiding behavior. Generally, light guiding relies on the total internal reflection principle. This means the light is confined to an optically thicker medium (the core) surrounded by an optically thinner medium (the cladding) through the total internal reflections at the core/cladding boundaries if the angle of the incident light *ϴ_inc_* at these boundaries ≥ the so-called critical angle *ϴ_c_*. 

The fiber-like components in CSs are of dimensions comparable to *λ_vacc_* (considered as the core), and, if embedded within a homogenous medium of lower refractive index (considered as the cladding), symmetric waveguide behavior is expected [77]. Although the complete analysis of the waveguiding behavior is left for future work, some related facts and observations are summarized here. Within the waveguide, the light propagates in discrete modes, and the number of supported modes depends on the ratio between its width *W_wg_* and *λ_sl_* (2**W_wg_*/*λ_sl_* = 2**W_wg_***n_sl_*/*λ_vacc_*) [77]. For instance, in our case (*W_wg_* = *W_M_* = *W_girdle_* = 0.184 µm, *n_sl_* = *n_v_* = 1.46), the cut-off *λ_vacc_* of the first mode ≈ 537 nm. This is changed in the CSs with changing *W_M,CS_*, e.g., CS_long,7_ (*W_M,CS_* = 0.334 µm), where the cut-off *λ_vacc_* of the first mode ≈ 975 nm. Nevertheless, the zero mode can be supported within the waveguide regardless of this ratio [77]. 

In the waveguide, the supported modes have a standing wave pattern associated with an evanescent field [78]. The confinement of a given mode inside the core, as well as the penetration depth of its evanescent field into the surrounding medium, is related to *W_wg_* and *∆n* [77]. In Appendix A, a standing wave pattern appears inside S_ana,M_ (*W_sl_ = W_M_* = 0.184 µm), limited by its height *h_sl_* = *h_M_* = 0.58 µm, across the whole studied *λ_vacc_* range. By increasing the height *h_sl_* of S_ana,M_ to 2 µm, the standing wave pattern becomes more evident (Figure 6a). This pattern changes by rotating S_ana,M_ and, thus, changing the *ϴ_inc_*, as in the case of tilted S_ana,M_ (Appendix A), similar to the mantle in the CSs, with simultaneous changes in its associated diffraction fringes.

As shown in Figure 6b, by approaching the structure of a cross-section across the complete frustule via adding four girdle bands (*H_girdle_*, *W_girdle_*)—two adjacent to the epivalve and the other two adjacent to the hypovalve with a step difference equaling *W_girdle_* on the y-axis and a spacing *d_girdle_* = 10 nm on the x-axis—further modulation in edge diffraction fringes is observed. This includes an additional spatial delay, especially the inside fringes (indicated by the blue arrows in Figure 6b,c). 

In most simulations carried out here, the incident wave falls onto the CSs’ external faces. Alternatively, as shown in Appendix A, by rotating the CSs 180°, the incoming wave falls onto the CSs’ internal faces, reaching the mantle first. As there are two opposite curved edges for each CS, the standing wave extends from each curved edge to the rest of the CS, interferes inside it, and seemingly causes guided-mode resonance (GMR)-like behavior, occurring across the whole studied *λ_vacc_* range in air and appearing in all CSs. This is accompanied by an evanescence with simultaneous changes to the transmittance and reflectance interference patterns (Appendix A). The verification and analysis of this mantle-coupled GMR-like behavior—distinct from grid-coupled GMR (vide infra)—are left for future work.

#### 3.3.4. Diffraction-Driven Focusing in the near Field

As explained, a consequence of reducing the *L_Sl_* of the thin-slab element is the arising of a distinct interference pattern with bright (red) spots alternating with dark (blue) spots in the transmittance region and influenced by the curved edges of the CSs and their waveguiding behavior (Figure 5). The bright (i.e., focusing) spots are distinctly visualized in Figure 7a and are quite similar to the pattern of the transmitted light through an aperture (see Figure 1 in [79]). The intensity of these spots, as well as their size, depends on the diffraction fringes they involve. This could be why the highest intense spots appear at the right-hand side of the interference pattern associated with the more intense diffraction fringes inside the transmittance region (e.g., Figure 7a). Further, adding more edges to the CSs, for instance, in CS_long,1_, where two raphe slits (representing four additional edges) are introduced, or in CS_ver,1_ and CS_ver,2_, where an increased thickness (associated with the nodule) is introduced at the center of CSs, leads to the presence of additional point sources—secondary wavelets—interfering with the transmitted wave. This leads to splitting the associated CS’s interference pattern into two separate but smaller patches of these spots, as can be seen in CS_long,1_ (Appendix A) and CS_ver,2_ (Appendix A).

Three parameters can be considered to study these focusing spots, as illustrated in Figure 7a: the distance between the CS and a given focusing spot *Z_f_*, its length *L_f_*, and strength *E_Norm,f_*. With increasing *λ_vacc_*, all these spots move toward the CS, decreasing *Z_f_* (Figure 7c) and *L_f_* (Figure 7d) and fading in its intensity (Figure 7b). In this case, *E_Norm,f_* is affected by two factors: the reduction of the strength of the secondary wavelets produced from the edges (as expected from the Fresnel–Kirchhoff integral) and the change caused by thin-film interference affecting the transmittance. In Figure 7b, the correlation between the transmittance intensity, as calculated from thin-film interference theory (vide supra), and the *E^2^_Norm,f_* of the focusing spots is—to some extent—evident. All spots follow a similar trend (see, e.g., Appendix A), even the spots that result from the interference with the so-called photonic jet, e.g., spots 1 and 3 in Appendix A.

Increasing *n_m_* (*n_v_* = 1.46), and, thus, decreasing *Δn*, leads to a significant increase in *Z_f_* (Appendix A) and *L_f_* and a simultaneous fading in *E_Norm,f_*. This is not the case for changing *n_v_* (*n_m_* = 1.00), as the changes in *Z_f_* and *L_f_* are negligible. 

This phenomenon is directly relevant to the distance between the edges; therefore, increasing the *L_Sl_* of S_anal,long5_, and, thus, the distance between the generated secondary wavelets, dramatically increases *Z_f_* (Appendix A) and *L_f_* while slightly decreasing *E_Norm,f_*. For a much larger thin-slab component, as the edge diffraction becomes insignificant again, this phenomenon becomes neglectable. Changing *D_Sl_* (from 50 to 400 nm) of S_anal,long5_ at *λ_vacc_* 330 nm leads to negligible changes in *Z_f_* and *L_f_*, while *E_Norm,f_* increases from 1.04 V/m (*D_Sl_* = 50 nm) to 1.38 V/m (*D_Sl_* = 400 nm).

From the observed edge diffraction and waveguiding behavior, it can be concluded that changing the edge geometry can dramatically affect these focusing spots; for instance, the increased *W_M,CS_* in CS_long,7_ might contribute to its increased *E^2^_Norm,f_* if compared to that of CS_long,5_ (Figure 7b). Studying the effects associated with edge geometry is left for future work.

#### 3.3.5. Photonic Jet

A photonic jet (PJ, also known as a photonic nanojet) is observed as a distinct focusing phenomenon in some CSs (all vertical CSs, CS_long,1_, CS_long,2_, and CS_long,3_), linked to the presence of an increased thickness in the CS either by introducing the nodule zone (e.g., Figure 8) or the sternum (e.g., Appendix A), which significantly affects the associated interference pattern. By separating the related optical component (e.g., CS_ver,2/nodule_ and CS_long,3/nodule_ in Figure 8c,d, respectively), their correlation to this phenomenon becomes evident. This type of focusing occurs when a plane wave is incident on a microscopic dielectric object of comparable size to *λ_vacc_* (for example, microspheres of radius ≈ 1–30 *λ_vacc_* [80,81,82,83]). PJ has been extensively studied during the last decade, both numerically and experimentally, for a wide range of dielectric microparticles of symmetric and asymmetric geometries, especially microspheres and microcylinders [80,82,84]. The mechanism of PJ focusing can be explained via the Mie scattering theory [82] as well as near-field diffraction approximations [79]. Our simulation results indicate the correlation of edge diffraction to this phenomenon, similar to the diffraction-driven focusing spots. 

It is expected from the previous work on PJ generation by artificial structures that the characteristic features of the PJ beam (i.a., the position, length, waist size, and maximum intensity) can be changed with changing parameters such as *n_v_*, *n_m_*, and incident *λ_vacc_*, as well as the structure size [79,80]. As illustrated in Figure 8e, the maximum intensity of the PJ (*E_Norm,PJ_*) generated by CS_long,3/nodule_ decreases exponentially with increasing *λ_vacc_* combined with a slight decrease in its distance from CS_long,3/nodule_ and an increase in its waist size (Appendix A). Furthermore, changing the refractive index contrast *Δn* by either varying *n_v_* (*n_m_* = 1) or *n_m_* (*n_v_* =1.46), but keeping *n_v_* > *n_m_*, leads to similar changes in the PJ parameters (Appendix A vs. Appendix A). Where the *E_Norm,PJ_* increases, its waist size decreases, and the distance to CS_long,3/nodule_ slightly decreases with increasing *Δn*. Similar trends were observed by Salhi et al. [79] that the PJ intensity decreases while its waist size increases with increasing *λ_vacc_* (see Figure 6 in [79]) or reducing *Δn* (see Figure 5 in [79]). 

Interestingly, an intense beam observed inside the simulation domain for a thin slab of *L_sl_* = 5 µm, *D_sl_* = *D_v_*, and *n_sl_* = *n_v_* appears as a PJ emerging after the diffraction-driven focusing spots within the transmittance region. By further reducing *L_sl_*, the generation of PJs occurs very close to the surface of the rectangular slab (Appendix A), leaving no space for the formation of the diffraction-driven focusing spots, which could also be the case for PJ generation by the nodule or the sternum of small dimensions. This means that even the CSs without the increased thickness at the middle can generate a pronounced PJ if their length, thickness, and edge geometry enable it. A pronounced PJ is also observed in the transmittance region beyond the 2D CSs of the intact frustule, such as in the examples shown in Figure 11 (vide infra).

#### 3.3.6. Diffraction Grating Behavior in the near Field: The Talbot Effect

The grid-like element in some CSs—as it is composed of optically transparent material—considers a transmission grating [74,85], where the diffraction orders mainly appear in the transmittance. As our simulations provide high-resolution information in the near field, we thus expect to obtain a periodic interference pattern matching the so-called Talbot effect [86,87] in the transmittance associated with the grid-like element. For a one-dimensional grid, in our 2D simulation domain, this effect leads to an interference pattern consisting of linear arrays of periodically arranged bright fringes (i.e., high-intensity fringes)—on the y-axis—alternating with dark fringes, where the two consecutive bright fringes have a spacing equal to the grid period *d*. Each linear array of bright fringes represents an image copy of the grating repeated at fixed distances—on the x-axis—equal to “Talbot length *Z_T_*” [86] alternating with other secondary copies occurring at *Z_T_*/2 with a lateral shift—on the y-axis—equaling *d*/2. The Talbot length *Z_T_* can be calculated according to Equation (1), specified for the small-period diffraction gratings [86].
(1)$$z_{T}=\;\frac{\lambda_{vacc}}{n_{m}(1-\sqrt{1-{\left(\frac{\lambda_{vacc}}{n_{m}*\;d}\right)}^{2})}}$$

This phenomenon occurs due to the interference of the wavefronts of different orders of diffraction in the near field, where the occurrence of additional sub-images at fractions of *Z_T_* depends on the number of diffraction orders [87]. This can explain the intense interference pattern which dominates the transmittance region of CS_long,3_, CS_long,4_, and CS_long,6_ at a range of wavelengths (e.g., CS_long,4_ in Figure 9a), which interrupts the edge diffraction pattern and the associated diffraction-driven focusing. The results show that this pattern occurs only where at least ± 1^st^ orders of diffraction are present in the transmittance beside the 0^th^ order. The possible number of diffraction orders can be calculated under normal incidence using the grating Equation (2). By solving this equation for the CSs, the presence of ± 1^st^ orders of diffraction is possible only at *λ_vacc_* < *d*n_m_* (*d = d_str_* or *d_a_* in the case of longitudinal or vertical CSs, respectively), while the ± 2^nd^ orders of diffraction can occur only at *λ_vacc_* < (*d*n_m_*)*/2*, which falls mostly outside the studied *λ_vacc_* range. This explains the observation of Talbot pattern in CS_long,3_ (*d_str_* = 490 nm), CS_long,4_ (*d_str_* = 500 nm), and CS_long,6_ (*d_str_* = 565 nm) at *λ_vacc_* < 490 nm, 500 nm, and 565 nm, respectively, in the air (*n_a_* = 1.00) and *λ_vacc_* < 652 nm, 665 nm, and 752 nm, respectively, in water (*n_w_* = 1.33). However, it cannot be obtained in CS_ver,1_ (*d_a_* = 214 nm) (Appendix A) at the studied *λ_vacc_* range either in air or water, which is confirmed for an equivalent analytical grid without nodule G_ana,ver1_ (Appendix A).
(2)$$sin\;\theta_{m}=m\;\left(\frac{\lambda_{vacc}}{n_{m}*\;d}\right)$$

*θ_m_* is the diffraction angle for a given diffraction order, and *m* is an integer that refers to the diffraction order (0, 1, 2, etc.).

The *Z_T_* dependency on *λ_vacc_* and *∆n* is investigated utilizing CS_long,4_ (*d_str_* = 500 nm), while the *Z_T_* dependency on structural parameters is further studied on an equivalent analytical grid G_ana,long4_ of rectangular grooves. The distinct difference between G_ana,long4_ (Appendix A) and CS_long,4_ (Figure 9a) is the presence of the mantle as well as the lens-like grid units in CS_long,4_, where the curved surface of the lens occurs at the internal valve face and has a depth of 70 nm out of the total valve thickness *D_v_* = 170 nm (Figure 2 and Appendix A). Despite this, no significant change is observed on *Z_T_* between the actual and analytical grid of the same *d* (Figure 9a vs. Appendix A), except for the increased deformation of the fringes in the case of CS_long,4_. In all cases, the extracted *Z_T_* has a slight deviation compared to theoretical predictions (Figure 9 and Appendix A) due to the Talbot fringes’ deformation caused by the edge diffraction, a consequence of the limited number of grid units [88]. This deformation leads to uncertainty while defining the Talbot planes (indicated by black dashed lines in Figure 9a and Appendix A). Thus, the extraction of *Z_T_* is based on the average, on the x-axis, accompanied by a deviation representing the uncertainty in the position of the Talbot planes. The extraction of *Z_T_* from CS_long,3_ or CS_long,6_ faces more difficulties related to the presence of the nodule zone or the stronger edge effect, respectively.

As illustrated in Figure 9, the Talbot length *Z_T_* associated with CS_long,4_ increases with decreasing *λ_vacc_* or increasing *n_m_* (*n_v_* = 1.46), which perfectly matches the theoretical expectation from Equation (1) and is correlated with the changes in *ϴ_m_*, which correspond to the ± 1 diffraction orders that can be calculated from Equation (2). While a dramatic change in *Z_T_* can be induced by changing the grid period *d* of G_ana,long4_ (Figure 9g). In contrast, neither changing the fill factor *ff* (i.e., the grid unit width/*d*) (Figure 9g), the number of grid units (Appendix A), nor the thickness *D* of G_ana,long4_ (Appendix A) changes *Z_T_*. The same conclusion is obtained for changing the *n_v_* (*n_m_* = 1) of CS_long,4_ (Appendix A). 

Interestingly, no significant *λ_vacc_* dependency is noticed for the maximum strength of the Talbot bright fringes *E_Norm,Talbot_* of CS_long,4_, which vary within the range of 1.36–1.56 V/m (with errors up to ± 0.07 V/m). Generally, the *E_Norm,Talbot_* shows dependencies on *∆n*, *D*, and *ff* but is independent of the grid unit number. Decreasing *∆n* (from 0.46 to 0.06) either by changing *n_v_* or *n_m_* decreases the *E_Norm,Talbot_* of CS_long,4_ with the same magnitude, e.g., *E_Norm,Talbot_* reaches 1.14 ± 0.02 V/m in water (*n_v_* = 1.46) at *λ_vacc_* 350 nm. Additionally, increasing the *D* of G_ana,long4_ (from 50 nm to 400 nm) increases *E_Norm,Talbot_* (from 1.11 ± 0.01 V/m to 1.71 ± 0.05 V/m), while decreasing *ff* increases the transmitted light through the areolae and, thus, enhances *E_Norm,Talbot_* significantly.

#### 3.3.7. Guided-Mode Resonance

For a specific range of wavelengths, the grid-like element exhibits another phenomenon, guided-mode resonance (GMR), which is widely reported for dielectric resonant gratings and photonic crystals of period *d* comparable to *λ_vacc_* [89,90,91]. At a specific combination of various parameters related to the incident light, including *λ_vacc_*, *ϴ_inc_*, and polarization, and the grid, including thickness *D*, pore spacing *d*, filling factor *ff*, and its material refractive index *n*, the grid-coupled GMR can be obtained [90,92,93,94]. It occurs when the grid (considered as an inhomogeneous waveguide) couples the incident wavefront into guided modes, which cannot be sustained within, and leak out [92] to interfere constructively or destructively with the reflectance or transmittance, respectively, leading to the characteristic spectrum of GMR [90]. During GMR, a standing wave pattern is expected inside the grid-like element (e.g., Figure 10), with intense nodes reaching the maximum at the middle of the grid and decaying towards the edges with a simultaneous evanescence in the proximity of the grid surface.

In the CSs with a grid-like element (CS_long,3_, CS_long,4_, and CS_long,6_), the *E_Norm,T_* dramatically drops at specific ranges of *λ_vacc_*, reaching a minimum value at *λ_vacc,GMR_* before returning to its normal limits with a simultaneous increase in *E_Norm,R_* that matches the expected behavior of GMR. Nevertheless, the extraction of *E_Norm,T_* or *E_Norm,R_* employing the same method described in Appendix A is complicated due to the finite size of the grid as well as the presence of other overlayed phenomena, such as the Talbot effect in the case of the first mode. Alternatively, to study the GMR behavior and to define *λ_vacc,GMR_*, very fine sweeps—down to 1 nm steps—are applied concurrently with observing the *E_Norm_* strength inside the waveguide, which reaches its maximum *E_Norm, GMR_* at *λ_vacc,GMR_*. The first mode of GMR is observed at *λ_vacc,GMR_* 295 nm, 303 nm, and 339 nm for CS_long,3_, CS_long,4_, and CS_long,6_, respectively, while the zero modes are obtained at *λ_vacc,GMR_* 523 nm, 553 nm, and 613nm, respectively, in the air. The presence of a defect in CS_long,4*_ causes a slight shift in *λ_vacc,GMR_* to occur at 559 nm and 307 nm for the zero (Figure 10c) and first modes (Figure 10d), respectively, where the maximum *E_Norm, GMR_* is associated with the defect rather than the geometrical center of the grid-like element. Further, by increasing *ϴ_inc_*, a splitting in the modes is observed (e.g., zero mode of CS_long,4_ in Appendix A) combined with a decrease in *E_Norm, GMR_* at *λ_vacc,GMR_*. Such behavior is expected (e.g., see Figure 3 in [92]).

Moreover, increasing *∆n* by increasing *n_v_* (*n_m_* = 1.00) leads to a red shift in *λ_vacc,GMR_* of the zero mode of CS_long,4_ (Appendix A) and a simultaneous increase in *E_Norm, GMR_* from 1.50 V/m (*n_v_* 1.13) to 8.04 V/m (*n_v_* 1.80). In contrast, increasing *∆n* with decreasing *n_m_* (*n_v_*= 1.46) leads to a blue shift in *λ_vacc,GMR_* (Appendix A) associated with an increase in *E_Norm, GMR_* from 1.38 V/m (*n_m_* 1.33) to 6.31 V/m (*n_m_* 1.00).

Furthermore, the influence of the structural parameters on GMR is studied on G_ana,long4_, where only slight shifts occur for *λ_vacc,GMR_* (zero and first modes at 556 nm and 303 nm, respectively) compared to *λ_vacc,GMR_* of CS_long,4_. Although the further analysis is focused on the zero mode, similar trends are expected for the first mode. Increasing the grid unit number of G_ana,long4_ leads to slight red shifts in *λ_vacc,GMR_*, which is more significant for grid units < 20 (Appendix A), combined with a dramatic increase in *E_Norm, GMR_*, reaching 11.70 V/m for 42 grid units. By increasing the *D* of G_ana,long4_, a red shift in *λ_vacc,GMR_* occurs (Appendix A), accompanied by less dramatic changes in *E_Norm, GMR_*, with a maximum (6.14 V/m) at *D* = 250 nm. Finally, increasing the *d* of G_ana,long4_ (at the same *ff*) causes a dramatic red shift in *λ_vacc,GMR_* (Appendix A) and a simultaneous decrease in *E_Norm, GMR_*. This explains the *λ_vacc,GMR_* shift observed for CS_long,3_ and CS_long,6_ associated mainly with the change in *d_str_*. Decreasing the fill factor *ff* (from 0.8 to 0.4) causes a blue shift in *λ_vacc,GMR_* (Appendix A) combined with a decrease in *E_Norm, GMR_*.

### 3.4. The Case of 2D Cross-Sections of a Complete Frustule Immersed in Water

As previously illustrated, adding the girdle bands spatially delays the inner edge diffraction fringes, probably due to the delay of the interference between the secondary wavelets generated at the edges—due to waveguiding behavior—and the transmitted wavefront through the CS (Figure 6b,c). By further adding the hypovalve—to form a CS in the complete frustule—almost all inner edge diffraction fringes, and, as a consequence, the diffraction-driven focusing spots, are removed from inside the frustule to appear beyond the hypovalve, as can be clearly seen in the CS_long5,frustule_ (Figure 11a), and, therefore, become irrelevant to photosynthesis. At the same time, the thin-film interference still affects the area inside the frustule, where the edge diffraction effect is diminished.

Moreover, the generated PJ by the nodule, or the sternum, is still observed inside the frustule (e.g., CS_long3,frustule_ in Figure 11b). Interestingly, the direction of this PJ follows the *θ_inc_* of the incident wave (Appendix A). This feature is confirmed in CS_long,3/nodule_ (Appendix A) but with more stability inside the complete frustule, which extends to a larger *θ_inc_* (Appendix A). This also leads to the generation of a PJ beyond the hypovalve associated with the nodule integrated into the hypovalve of CS_long3,frustule_ (Figure 11b). Another stronger PJ appears beyond the hypovalve observed in all CSs of the complete frustule associated with the frustule edge diffraction (black arrows in Figure 11). 

Furthermore, the Talbot fringes—associated with the grid-like element—remain inside the frustule but also appear beyond the hypovalve, as shown in Figure 9d, while minimizing the edge effect inside the frustule weakens the lateral deformation of the inside Talbot fringes. This is clear when comparing CS_long4,frustule_ (Figure 9d) to CS_long,4_ (Figure 9b), given that both are in the water at *λ_vacc_* 350 nm. 

Finally, there are no shifts in *λ_vacc,GMR_* of the zero mode observed for the epivalve in CS_long4,frustule_ either in air (Appendix A) or in water (Appendix A), while the standing wave extends from the epivalve to the hypovalve through the mantle and the girdle bands. Underwater, the transmittance beyond the frustule does not attenuate during GMR when compared to air (Appendix A vs. Appendix A).

## 4. Discussion

The comprehensive structural analysis of GP frustules utilizing FIB-SEM analysis, in addition to the regular SEM, offers high-resolution structural details crucial for predicting their light modulation abilities. While statistical analysis suggests that some critical structural parameters, such as the valve thickness *D_v_* and striae spacing *d_str_*, are reproducible between different valves within the studied culture, this likely means they might be built and optimized by the living cells on purpose to contribute to potential photobiological roles. In future work, our proposed FIB-SEM workflow could—using the cryo-fixation of complete cells—provide detailed information about the exact location of, e.g., chloroplasts with respect to the frustule, or the existence of other materials and layers that could influence the refractive index contrast in the vicinity of the frustule and, thus, their interaction with light. This method enables data generation as close to the real state as possible, showing the cell’s interior features without significant alteration. 

Through extensive numerical analysis, the ability of the GP valve, as well as the complete frustule, to modulate light in the near field is demonstrated and explained. Using 2D CSs and disassembling distinct optical components enable understanding of the complex interference patterns (e.g., Appendix A) and the finding of their correlation to the known optical phenomena in the near field and micro-optics. Further, using analytical models allows the determination of the significance of structural parameters to the observed phenomena but also enables future prediction of the light modulation capabilities of other unstudied pennate species. 

At that point, it should be noted that, although the numerical analysis of 2D CSs gives a deeper analytical understanding of the involved optical phenomena and the general trends, there are some limitations. The actual shape and light intensities, e.g., those of PJ and Talbot fringes, occurring in 3D cannot be obtained from 2D simulations. Moreover, in the case of grid-coupled GMR, *λ_vacc,GMR_* and *E_Norm,GMR_* are expected to be shifted when transferring from the 2D to 3D situation. Such information can be obtained from 3D simulations, which are left for a follow-up study. 

In the following subsections, we—in light of the obtained results and recalled previous work—try to predict near-field light modulation by an intact three-dimensional GP valve, its potential for applications (4.1), and, further, the hypothetic photobiological relevance of their frustules (4.2). 

### 4.1. The Light Modulation by GP Valve: The Competing Phenomena and Potential for Applications 

The light modulation in 3D, with the presence of all integrated optical components in such small-size valves, is expected to give a more complicated, probably more intense, interference pattern but will also show how different physical phenomena are competing. 

The GP valve, like the valve of many other pennates, consists of a single optically thin, porous silica layer of a thickness *D_v_* ≤ the visible light wavelengths; thus, thin-film interference is expected. This is distinct from the multilayer structure of some centric and pennate valves associated with the presence of loculated areolae [11], which probably leads to multilayer interference behavior. To the best of our knowledge, this phenomenon has not been studied before for pennate valves. In contrast, the interference fringes have been witnessed in the reflectance spectrum of a centric valve of *Coscinodiscus wailesii* with a multilayer structure (see Figure 2 in [95]). In GP, although the thin-film interference associated with the thin-slab element is disturbed by (I) the edge diffraction and (II) the presence of integrated optical components (including the 1D grid-like and lens-like components) which cover the valve area, as can be concluded from Figure 1 and Appendix A, the intensity of the reflected and transmitted light is expected to affect by this phenomenon significantly. This, in turn, is expected to affect the strength of the final interference pattern, which is evident in the case of the diffraction-driven focusing spots’ intensity *E^2^_Norm,f_* (Figure 7b). This means that, under normal incidence, the thickness *D_v_*, as well as *n_v_*, may be crucial for determining the *λ_vacc_*-dependent reflectance/transmittance ratio of the valves. 

Moreover, unlike in large valves, such as those of *Coscinodiscus* spp., where the edge diffraction is less significant [96], in finite-size valves, the edge diffraction significantly contributes to the light interference pattern. A similar conclusion was obtained for the valve of pennate diatom *Ctenophora pulchella* of narrow width (see Figure 2 in [29]). The contribution of thickened areas within the valve (such as the nodule in GP) to the obtained interference pattern suggests that, even in the case of large valves with complex ultrastructures, such as the valves of *Arachnoidiscus* spp. (see Figure 3 in [47]), the diffraction from those additional edges is expected to play a significant role in their light modulation behavior.

Across all the studied *λ_vacc_*, the PJ is expected to be dominant at the apical axis associated with the presence of the nodule and the sternum, which might be the case for all biraphid pennates and can be considered a special case of diffraction-driven focusing. The PJ associated with the nodule (the maximum *D*) is expected to be higher in intensity and interrupts the interference pattern, as shown in Figure 8. This PJ is similar to the focusing beam observed by De Tommasi et al. [29]. The intense light focalization might give the PJ—especially that associated with the nodule—potential in several applications to, for instance, enhance the resolution of optical microscopy, reach super-resolution imaging, enhance Raman scattering, improve fluorescence spectroscopy, enable subwavelength photolithography, and enhance the optical absorption in optoelectronic devices [80,81]. Additionally, by changing *∆n*, either via changing *n_m_* or *n_v_*, the features of the PJs can be tailored, as shown in (Appendix A), which could be interesting for future applications. 

Furthermore, inside the transmittance region of GP valve, the Talbot interference pattern is expected to dominate at a range of wavelengths, where additional diffraction orders (± 1^st^ orders) besides the 0^th^ order can be present. For a clean GP valve immersed in air, the normal incident light of *λ_vacc_* < 500 nm is expected to generate Talbot fringes, representing an image copy of the 1D grid-like structure, which consists of the valve’s costae alternating with striae of increasing spacing (*d_str_*) toward the edges (Figure 1). For *λ_vacc_* between 500–565 nm, only a smaller part of the grid (closer to the edges) might be able to diffract the light into ± 1 orders of diffraction and, thus, might not generate a clear Talbot pattern. In general, the Talbot fringes are expected to be distorted near the valve edges, influenced by the edge diffraction and, near its apical axis, influenced by the PJ (Figure 8b). Although the Talbot effect is well known in the near-field optics of diffraction gratings [86,97], it is often not mentioned—apart from by De Stefano et al. [28]—or analyzed in diatom-related studies. On the contrary, the analysis of far-field diffraction grating behavior is more frequent, as in [38], which, unlike near-field behavior, is not directly relevant to photobiology or most applications. Recently, the Talbot phenomenon has been utilized in several applications, for instance, in fluorescence Talbot microscopy [98,99], displacement Talbot lithography [100,101], and image sensors [102]. It should be noted that the Talbot pattern produced by GP valves may not be appropriate for such applications if compared to the valves of larger-size pennate species such as *Nitzschia* and *Hantzschia* spp. (see Round et al. [11]), where the grid-like component has an almost fixed *d_str_* and is not interrupted by a sternum or nodule at the valve apical axis. 

Additionally, at a narrower range of wavelengths, the 1D grid-like element is expected to initiate grid-coupled GMR, where the transmittance drops dramatically and, thus, affects the intensity of the Talbot fringes at the first mode or diffraction-driven focusing spots at the zero mode. A consequence of changing *d_str_* is the changing of the associated *λ_vacc_* ranges toward the edges (as can be concluded from Appendix A) with a simultaneous decrease in the efficiency of the grid-like element—due to the decrease in grid units from CS_long,3_ to CS_long,6_—to couple the incident waves into guided modes. The GMR has previously been reported in diatom valves associated with the periodic pore arrays of some species, such as *Pinnularia* spp. [36]. Most previous studies focused on coupling GMR and surface plasmon resonances of the metallic nanoparticles or thin films to obtain efficient hybrid substrates for surface-enhanced Raman spectroscopy (SERS) [103,104]. In general, GMR can also be of interest for sensing applications and optoelectronic devices [89,94,105] associated with enhancing electromagnetic fields near the valve surface through the simultaneous evanescence field. Due to the GP valve’s finite size, and, thus, small grid unit number, the quality of GMR is expected to be reduced. Hence, larger pennate valves, especially of small areolae size (large *ff*), might be more appropriate for GMR-based applications. 

For the wavelengths away from the ability of the grid-like element to diffract the light or initiate GMR, the diffraction-driven focusing spots are expected to dominate the interference pattern in the near field, similar to this of a homogenous thin slab. As mentioned, the nodule zone is expected to further divide the interference pattern into separated parts (e.g., Appendix A). The observed behavior of these focusing spots is in good agreement with the focusing behavior reported in the previous work for the valves of *Coscinodiscus* spp. and *Arachinodiscus* sp. [28,30,47,106,107]. The distinct difference between the focusing spots observed in GP valve and those of large centric valves is their complete dependency on the edges (no pores needed). In contrast, the focusing spots generated by the large centric valves are more related to the superposition of the diffracted light from all pores’ edges [28,96]. This could be why the focusing spots remain inside the frustule of *Coscinodiscus* spp. [30], which is not the case in the GP frustule. It should be noted that the foramen pore diameters—and the period in between—in *Coscinodiscus* spp. valves (≈1–3 μm) are comparable to the size of the whole GP valve (≈4–7 μm). 

Finally, GP valves might be utilized in photonic applications based on and within the limits of the discussed competing phenomena. In such applications, the valves are often spread over a substrate to form a monolayer [42,108], where they occur in two configurations, either showing the external or internal face (Figure 12a or Figure 12b, respectively). Recent reports showed a degree of control over the valve orientation on the substrates that could be helpful for specific applications [108]. Most of our simulations are focused on the illumination of the external face. In this case, our results suggest that the thin-film interference, curved edge diffraction, and grid-coupled GMR collectively might lead to a *λ_vacc_*-dependent shielding effect of the valves that significantly attenuates shorter *λ_vacc_*, especially at higher *∆n* (e.g., in air), while adding the valve in that orientation to a substrate will lead to changes in, for instance, thin-film interference by adding a thin layer of medium trapped between the valve and substrate (Figure 12a). Although these presumed changes, the dense monolayer film might acquire a colligative shielding effect against a specific range of *λ_vacc_* similar to the colligative UV-shielding effect reported by Su et al. [109] for dense films of different centric valves.

On the other hand, when illuminating the internal valve face (Figure 12b), the incoming light might be coupled—across all optical wavelengths—into the valve through the mantle, initiating what we call mantle-coupled GMR-like behavior, as shown in Appendix A. This is less intense and has a different distribution pattern than the grid-coupled GMR but is expected to enhance the electromagnetic fields of all optical wavelengths in the proximity of the valve surface associated with the concurrent evanescence field. Despite this, the highest enhancement can be obtained at *λ_vacc,GMR_* of grid-coupled GMR, which is slightly shifted in this configuration. Further changes observed in this configuration (Appendix A) for the other optical phenomena are indeed interesting and can be explained according to the underlying physics but are left for future work. 

### 4.2. Hypothetic Photobiological Relevance of GP Frustules

As GP living cells live underwater, understanding the case of a complete frustule in water, illustrated in Section 3.4, is crucial for correlating the observed phenomena and the designated optical elements to hypothetical photobiological roles. Although there is significant reduction in the light modulation strength, all phenomena still occur and might presumably have photobiological functions. It should be noted that GP is a benthic species living near the bottom of the basins, where the blue–green spectral ranges dominate due to the strong absorption of the red and infra-red wavelengths. The phenomenon increases in significance with increasing of water column depth that the light penetrates before reaching the living cells [110]. The expected photobiological relevance might not only be limited to photosynthesis enhancement—by attenuating harmful radiation while maximizing absorption of PAR—but also might extend to perform putative signaling and sensing mechanisms [22,110]. 

In GP living cells, the chloroplast occurs adjacent to the valve—where the grid-like element (i.e., the striae and costae) covers most of its area—and to the fiber-like elements (i.e., mantle and girdle bands) [111]. This gives significant importance to the Talbot effect, grid-coupled GMR, and waveguiding behavior. In parallel, thin-film interference generally affects the transmittance inside the cell. In contrast, diffraction-driven focusing is not relevant to photosynthesis. PJ generation by the nodule and the sternum might also not be directly relevant to photosynthesis as it occurs along the apical axis within a narrow area compared to the chloroplast area. 

Moreover, a PJ is expected inside the frustule along the apical axis associated with the sternum and nodule but is of a higher strength at the nodule zone and shorter *λ_vacc_* (e.g., Figure 8). If we couple this fact with the observed *θ_inc_*-dependent direction of the PJs (Appendix A) and the fact that the nodule is in a relevant position to the nucleus, which has a richness of aureochrome and cryptochrome photoreceptors capable of sensing blue light [112]. Given that blue light, which is especially dominant in the benthos region, is crucial for many physiological processes of living cells, including the cell cycle [22]. All these facts together suggest a hypothetical PJ-based sensing mechanism for the light direction. This is inspired by a similar mechanism that has been suggested for the spherical cells of cyanobacterium *Synechocystis* sp. [113,114]. The illumination of these cells generates PJs at the shadow side, which are assumed to be perceived by a putative, well-distributed network of photoreceptors fixed on the plasma membrane, triggering a cellular signal transduction cascade ending by the flagella movement toward or outward from the light (see Figure 5 in [113]). However, unlike *Synechocystis* sp., the motility in raphid pennate diatoms, and, thus, GP, does not involve a flagellum, but rather gliding motility through the secretion of mucilage from their raphe slits [115,116]. 

## 5. Conclusions

Despite the 3D, complex ultrastructure of diatom frustules, the as-followed analysis logic flow could be promising for understanding their light modulation capabilities. Even tiny pennate frustules, such as our studied example, can modulate light effectively based on their ultrastructure, with strong dependencies on *λ_vacc_* and ∆*n*. This might not be surprising from physicists’ point of view, but it definitely is for diatomists and biologists. Our findings indicate that some optical phenomena are linked to the presence of integrated optical components, such as GMR and the Talbot effect associated with the grid-like structure, while other phenomena are more linked to the size parameter than the optical component itself, for instance, the generation of PJs. Studying the change of structural parameters—despite the ability of the studied GP strain to build many important parameters precisely—would help to predict the behavior of other GP strains and different species of the same genus but also different pennate genera of similar structure but different dimensions. Moreover, the separated GP clean valves might be valuable for some photonic applications, while their complete frustules might presumably be relevant to photobiological functions. Finally, our results could inspire ongoing research on the optics of diatom frustules, as well as artificial dielectric microstructures, in addition to their influences on photobiology. 

## Figures and Tables

**Figure 1 nanomaterials-13-00113-f001:**
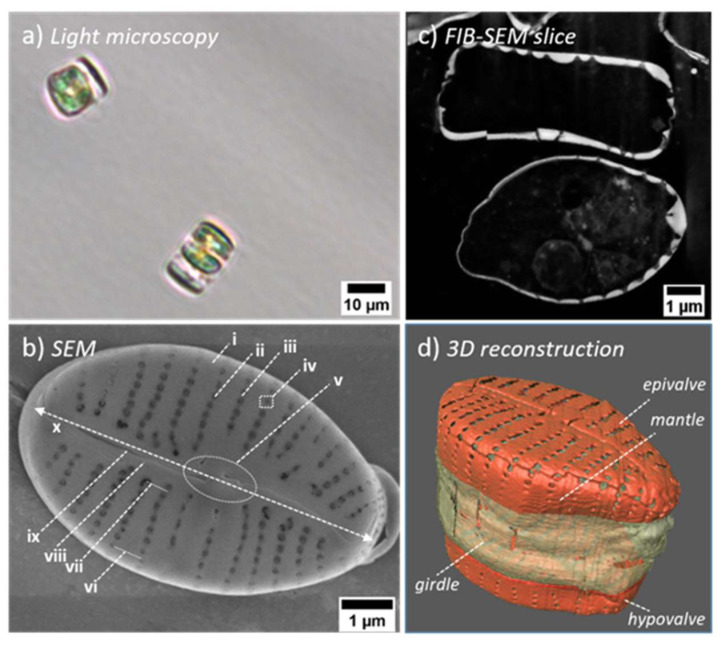
Light microscopy (**a**), SEM of the external valve view (**b**), FIB-SEM slice showing cross-sections of two individual cells (**c**), and 3D reconstruction of the frustule showing valves in red and girdle bands in white (**d**) of different *Gomphonema parvulum* cells. Some structural features are illustrated in (**b**), including: costae (i), striae (ii), a shortened stria (iii), punctate areola covered with a flab-like occlusion (iv), nodule zone (v), striae spacing max (vi)/min (vii), sternum (viii), raphe slit (ix), and the apical axis (x).

**Figure 2 nanomaterials-13-00113-f002:**
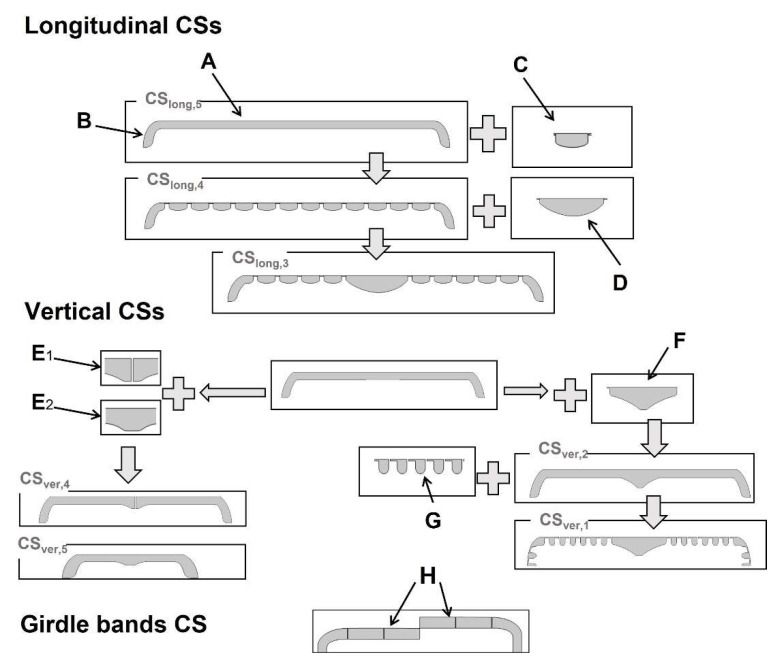
A schematic diagram illustrates the idea of “building” the CSs by adding different optical components. (**A**) A thin-slab element representing the valve face without areolae or other components, (**B**) a curved, fiber-like structure representing the mantle, (**C**) a grid unit of lens-like structure representing a costa, (**D**) lens-like structure representing the nodule in longitudinal CSs, (**E_1_** and **E_2_**) increased thickness representing the sternum with and without the raphe slit, respectively, (**F**) triangular-like structure representing the nodule in vertical CSs, (**G**) the grid-like structure in vertical CSs, representing the consecutive areolae within a stria, (**H**) four rectangular, fiber-like structures representing girdle bands.

**Figure 3 nanomaterials-13-00113-f003:**
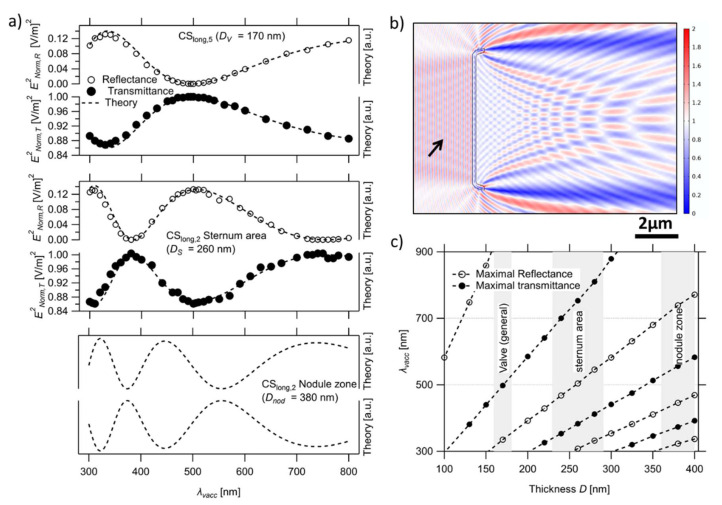
(**a**) The plots show the extracted *E^2^_Norm,R_* and *E^2^_Norm,T_* of CS_long,5_ and CS_long,2_ compared to the theoretical calculation for the studied *λ_vacc_* range. The estimated error in the extracted *E_Norm_* is up to ± 0.006 or ± 0.01 V/m in CS_long,5_ or CS_long,2_, respectively. (**b**) CS_long,5_ shows the maximum constructive interference at reflectance (*λ_vacc_* = 330 nm); the black arrow indicates the formation of standing waves between the reflected and the incident wavefronts. (**c**) The dependency of *λ_vacc_* positions of the constructive interference maxima on *D_sl_*. The grey-shaded areas in (**c**) show the significant error in the thickness of different valve parts (Table 1).

**Figure 4 nanomaterials-13-00113-f004:**
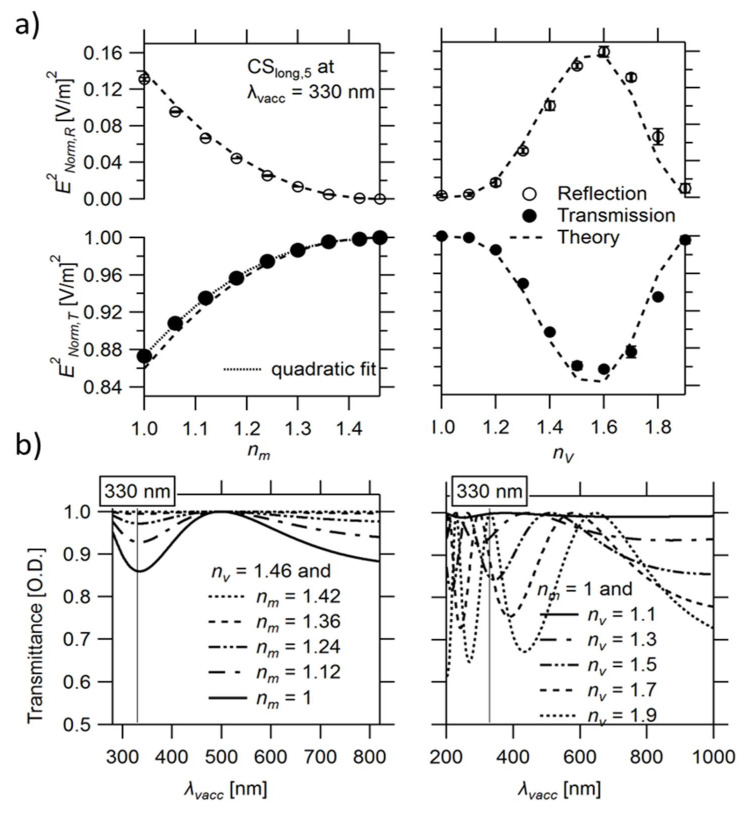
(**a**) The change of *E^2^_Norm_* (i.e., intensity) of the reflected and transmitted wave by CS_long,5_ at *λ_vacc_* = 330 nm vs. changing *n_m_* (left) or *n_v_* (right) compared to theoretical calculations. (**b**) The change of transmission spectra vs. changing *n_m_* (left) or *n_v_* (right), based on theoretical calculations.

**Figure 5 nanomaterials-13-00113-f005:**
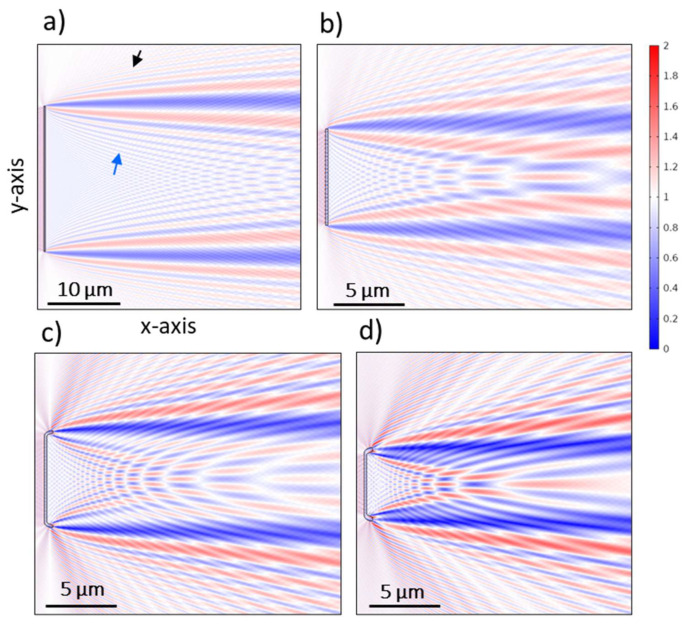
A thin slab of 20 µm length (**a**), S_ana,long5_ (**b**), CS_long,5_ (**c**), and CS_long,7_ (**d**) at *λ_vacc_* = 300 nm in the air. The black and blue arrows in (**a**) indicate the fringes outside and inside the transmittance region, respectively.

**Figure 6 nanomaterials-13-00113-f006:**
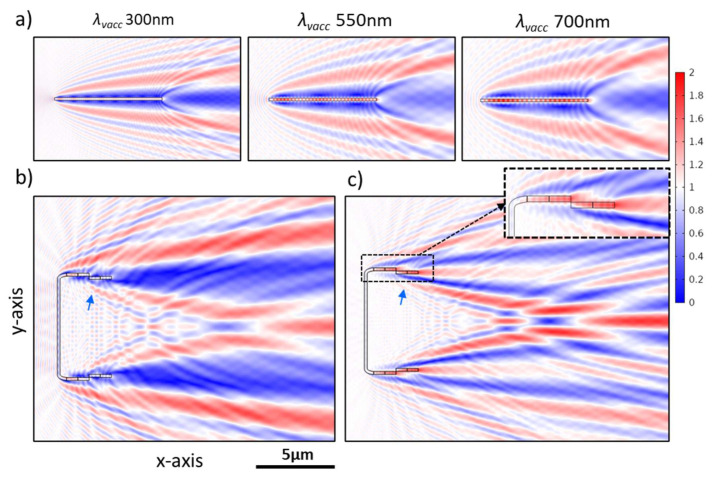
S_ana,M_ of height *h_sl_* = 2 µm in air (**a**), CS_long,5_ with four girdle bands in air (**b**) and in water (**c**) at *λ_vacc_* 550 nm. The zoom inset in (**c**) seemingly shows a coupling behavior in water that is not observed in air (**b**).

**Figure 7 nanomaterials-13-00113-f007:**
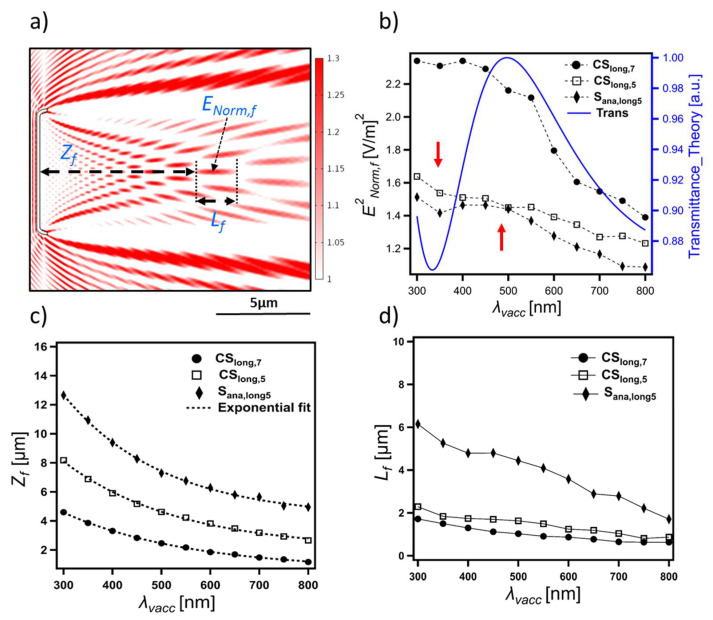
(**a**) The interference pattern of CS_long,5_ at *λ_vacc_* 300 nm in air and the focusing parameters of interest; the color code emphasizes the focusing spots. (**b**) The *λ_vacc_* dependency of the intensity *E^2^_Norm,f_* of selected spots (e.g., the spot illustrated in (**a**) for CS_long,5_) compared to the thin-film interference spectrum of the transmittance expected from theory (blue); the red arrows indicate places where the thin-film interference affects the intensity of the spots. The extracted *E_Norm,f_* has an error up to ± 0.02 V/m associated with the local variation in the strength of the selected spot. (**c**) The *λ_vacc_* dependency of *Z_f_* and (**d**) *L_f_* of the same selected spots in air.

**Figure 8 nanomaterials-13-00113-f008:**
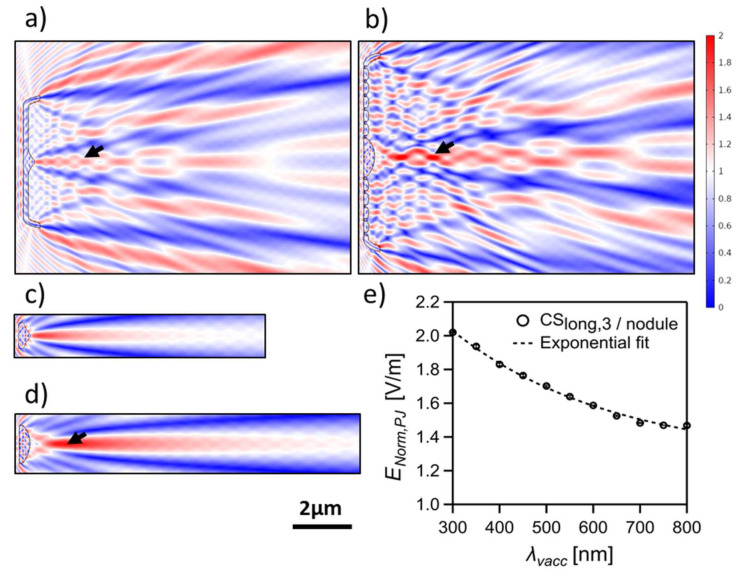
PJ generation in CS_ver,2_ (**a**) and CS_long,3_ (**b**) compared to the disassembled nodule zone CS_ver,2/nodule_ (**c**) and CS_long,3 /nodule_ (**d**) at *λ_vacc_* 350 nm in the air. (**e**) The plot shows the *λ_vacc_* dependency of *E_Norm,PJ_* of the PJ generated by CS_long,3 /nodule_ indicated by the black arrow in (**d**).

**Figure 9 nanomaterials-13-00113-f009:**
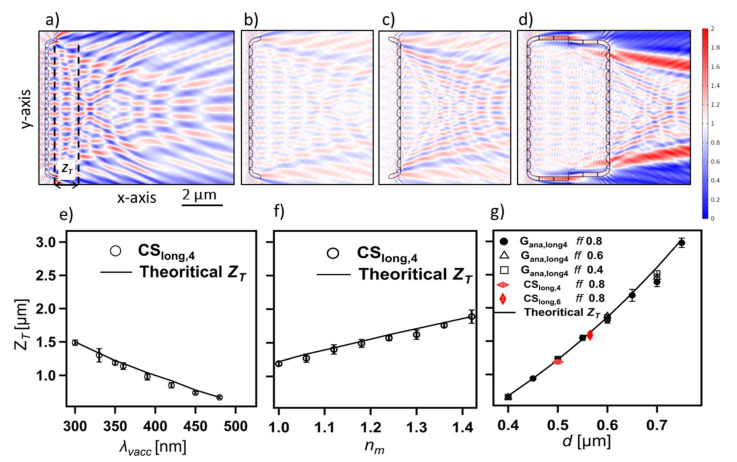
The interference pattern of CS_long,4_ displays the Talbot pattern dominating the transmittance region at *λ_vacc_* = 350 nm in air (**a**) and in water for the light incident on the external face (**b**) or the internal face (**c**). The distance between the two black, dashed lines (considered as the Talbot planes) in (**a**) is the so-called Talbot length (*Z_T_*). The Talbot pattern in CS_long,4, frustule_ at *λ_vacc_* = 350 nm immersed in water (**d**). The graphs illustrate the *Z_T_* dependency on *λ_vacc_* in CS_long,4_ in air (**e**) and on *n_m_* (*n_v_* = 1.46) at *λ_vacc_* = 350 nm (**f**). A graph shows the *Z_T_* dependency on *d* of G_ana,long4_ with different fill factors (*ff*) at *λ_vacc_* = 350 nm in air (**g**). The error bars in the graphs represent the uncertainty in the measured *Z_T_*.

**Figure 10 nanomaterials-13-00113-f010:**
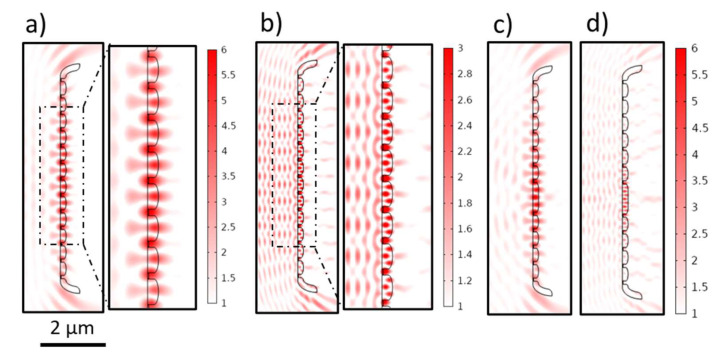
Grid-coupled GMR of CS_long,4_ at the maxima of zero (**a**) and first modes (**b**) and of CS_long,4*_ at the maxima of zero (**c**) and first modes (**d**) in air. The *E_Norm, GMR_* is 6.31 (**a**), 4.01 (**b**), 6.67 (**c**), and 5.17 (**d**) V/m at *λ_vacc,GMR_* 553 nm, 303 nm, 559 nm, and 307 nm, respectively. The color code emphasizes the *E_Norm_* enhancement in red (*E_Norm_* > 1 V/m) while both the *E_input_* and the *E_Norm_* reduction are in white (*E_Norm_* ≤ 1 V/m), not emphasized.

**Figure 11 nanomaterials-13-00113-f011:**
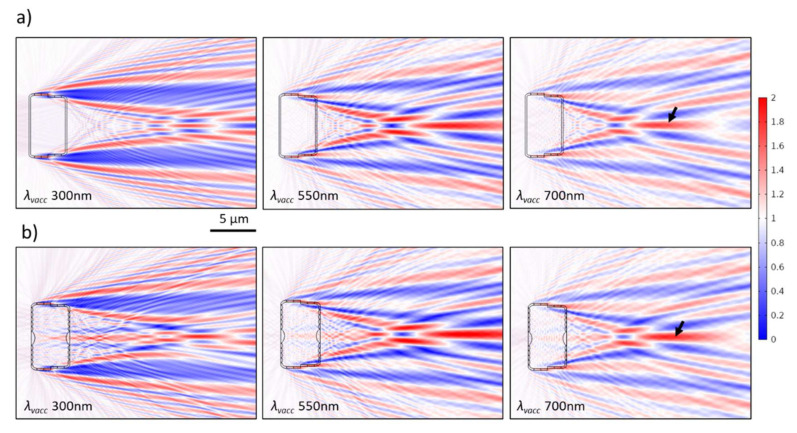
The interference pattern of CS_long5,frustule_ (**a**) and CS_long3,frustule_ (**b**), which represent a complete frustule consisting of CS_long,5_ or CS_long,3_, respectively, immersed in water at different *λ_vacc_*. With increasing *λ_vacc_*, the diffraction-driven focusing fringes that appear outside the CSs move toward them. The hypovalve (of the same structure) is slightly smaller than its epivalve. The black arrows indicate what seems to be a PJ.

**Figure 12 nanomaterials-13-00113-f012:**
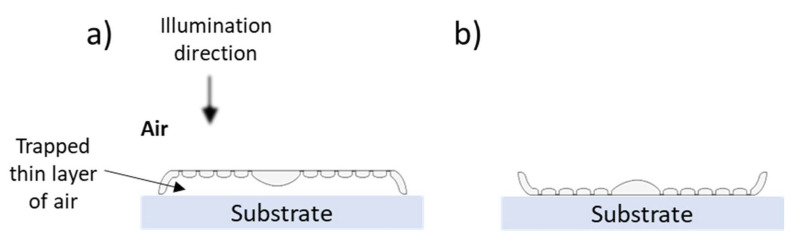
Schematic diagrams showing a GP valve (represented by CS_long,3_) laying on a substrate with the external (**a**) or internal face (**b**) fronting the light source.

**Table 1 nanomaterials-13-00113-t001:** Statistical analysis of the structural parameters of GP valves (weighted mean, internal and external errors—the bold print indicates the significant error—and its precision in %). The weighted mean is used for building the 3D model (see Appendix A).

Parameter	Description	*X_w_* (µm)	*dX_int_* (µm)	*dX_ext_* (µm)	Precision (%)
** *L_v_* **	Valve length	**7.1**	0.006	**0.2**	3
** *W_v_* **	Valve width	**4.59**	0.006	**0.07**	2
** *D_v_* **	Thickness of the valve	**0.17**	**0.01**	0.004	6
** *D_nod_* **	Thickness of the nodule zone	**0.38**	0.009	**0.02**	5
** *W_nod_* **	Width of the nodule zone	**0.86**	**0.03**	0.02	4
** *L_nod_* **	Length of the nodule zone	**1.568**	0.002	0.002	0.1
** *D_S_* **	Maximum thickness of the sternum except the nodule zone	**0.26**	**0.03**	0.01	**12**
⅟₂***W_s_***	Half-width of the sternum	**0.32**	**0.02**	0.01	5
** *L_ra_* **	Raphe slit length	**5.8**	0.006	**0.2**	3
** *W_ra_* **	Raphe slit width	**0.023**	0.0004	**0.004**	**17**
** *d_ra_* **	Raphe slit spacing at nodule zone	**0.54**	0.004	**0.06**	**11**
** *d_str,min/_* ** ** *d_str,max_* **	Striae spacing (center to center)	**0.49/0.57**	**0.03/0.02**	0.02/0.01	6/4
** *d_a_* **	Areolae spacing (center to center)	**0.214**	**0.008**	0.006	4
**2*r_a,ext_***	Areolae diameter (2x radius) external face	**0.100**	**0.007**	0.002	7
**2*r_a,int_***	Areolae diameter (2x radius) internal face	**0.15**	**0.01**	0.008	7
** *h_M_* **	Mantle height	**0.58**	0.004	**0.08**	**14**
** *W_M_* **	Mantle width	**0.184**	**0.009**	0.004	5
** *W_occ_* **	Width of the pore occlusion slit	**0.017**	0.002	0.002	**12**

## Data Availability

The data are available upon reasonable request from the corresponding authors.

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
