# Peer review of "Numerical Analysis of the Light Modulation by the Frustule of Gomphonema parvulum: The Role of Integrated Optical Components"

_nanomaterials, 2022, doi:10.3390/nano13010113_

Round 1

Reviewer 1 Report

In this work, the authors proposed a novel approach for analyzing the near-field light modulation by small pennate diatom frustules. Finite element approach was adopted for the wave propagation across selected 2D cross-sections in a statistically representative 3D model for the valve. Conclusions are obtained. It is an interesting work, and has certain practical significance. I suggest this manuscript can be accepted after minor revision. My specific comments are as follows,

-What is the advantages of the proposed method for analyzing the near-field light modulation? Please give some quantitative analysis by comparison with other methods.

-The introduction does not analyze enough the existing literature on the specific topic: there is no clear claim about what is original compared to what has been done by others on the same matter.

-Simulations are useful, but scientific contribution and novelty of this tests should be showed with more results. Results are not informative. Comparisons with others’ works are not enough.

-The reference part can be improved a lot by adding more recent optics researches, for example, Optical Fiber Technology, 42(1), pp 97-104, 2018.

-The previous work conducted by the authors is short of explanation, especially the experiments. Please add more researches done by the authors themselves on this topic.

Author Response

Dear Reviewer,

We would like to thank you very much for reviewing our work and your helpful comments.
Please see the answers in the attached file.

Best Regards,
Mohamed Ghobara

Reviewer 2 Report

In this manuscript, the authors developed a new method for analyzing the near-field light modulation by small pennate diatom frustules. The paper was well-prepared and written. The conclusions were fully supported by the results. I only had a minor comment. The font size in Figs. 3, 4, 7-10 was too small, especially Fig. 3. Please make the texts large.  

Author Response

(The authors gave the same response as above.)

Reviewer 3 Report

In this paper, the authors present an approach for analyzing the near-field light modulation by small pennate diatom frustules, utilizing the frustule of Gomphonema parvulum as a model. Numerical analysis was carried out for the wave propagation across selected 2D cross-sections in a statistically representative 3D model for the valve based on the finite element frequency domain method. The influence of wavelength of light (vacuum wavelengths from 300 to 800 nm), refractive index changes, as well as structural parameters on the light modulation was investigated and compared to theoretical predictions when possible. The results indicate complex interference patterns resulting from the overlay of different optical phenomena. This article is clear, concise, and suitable for the scope of the journal. Several small suggestions are supplied:
1.  Suggest the authors supply more detail in the sentence about the Talbot effect and characteristic Talbot length (ZT) in CSlong,4.
2. Suggest the authors supply more detail in the sentence about the λvacc dependency of light modulation by 2D CSs representing a complete frustule 1058 consisting of CSlong,5 (a) or CSlong,3 (b) immersed in water.
3. Suggest the authors check small typos.

Author Response

(The authors gave the same response as above.)

Reviewer 4 Report

The manuscript was well written and it is very original and interesting.  Three main remarks: i) the English should be revised by a native English speaker; ii) the Introduction and Materials&Methods sections are too long and should be improved by reducing the text and deleting less important information; iii) the conclusions should be written avoiding redundant information already discussed in the other sections and it should be focused only on the main results in relation to the aims of the study. A last suggestion: Results and Discussion sections are very interesting but they may be integrated into a single section to avoid repetitions.

Author Response

(The authors gave the same response as above.)

Reviewer 5 Report

This paper deals with light modulation of frustule. It is expected to contribute to the development of the field. It would be worthy of publication if the following points were corrected.

The authors show SEM photographs of the particles in Fig. 1b - at what stage in the production process does this microstructure arise? Also, the black dots are arranged in a straight line - why does this orderly arrangement occur?

Where is the process of creating these microparticles shown? You may be citing a previous paper, but please add the particle creation process in this paper as well (with a scheme diagram if possible) to help the reader's understanding.

The authors have shown a number of parameters in Table 1, but please provide a schematic diagram of the equipment and explain where the length of each variable is. How do these values affect, for example, the shape of the particles?

You have made various assessments in Figures 3-12, but what are the numerous parameters that appear here measuring for the particles? Also, please specify how these parameters change if the particle shape changes.

The paper as a whole provides a detailed analysis, but I felt that there was a lack of description of the fundamentals needed to understand these results.

It is well written. I consider it worthy of acceptance if the sections I commented on to the author can be corrected.

Author Response

(The authors gave the same response as above.)
